# The Effect of Hock Injury Laterality and Lameness on Lying Behaviors and Lying Laterality in Holstein Dairy Cows

**DOI:** 10.3390/ani7110086

**Published:** 2017-11-17

**Authors:** Nicole L. Eberhart, Peter D. Krawczel

**Affiliations:** Department of Animal Science, University of Tennessee, 2506 River Dr. 258 Brehm Animal Science Knoxville, Knoxville, TN 37996, USA; neberhar@vols.utk.edu

**Keywords:** laterality, lying behavior, cow, hock injury, lameness

## Abstract

**Simple Summary:**

Dairy cattle may experience discomfort in a myriad of ways throughout their life cycle, particularly when sustaining hock injuries or suboptimal locomotion. Lactating dairy cattle divide their lying time equally between left and right sides; however, discomfort experienced during pregnancy or following cannulation can cause a shift in the normal lying laterality. The objective of this study was to determine the effect of hock injuries and lameness on the lying behaviors of dairy cattle, particularly lying laterality. Lying laterality did not differ from the expected 50% (left side lying time) in cattle with hock injuries, lameness, or both. The current results suggest that lying laterality does not differ between varying levels of hock injury or lameness severity. Going forward, further research could determine if lying laterality shifts over the course of the animal developing a hock injury or lameness.

**Abstract:**

Lactating dairy cattle divide their lying equally between their left side and their right side. However, discomfort, such as pregnancy and cannulation, can cause a cow to shift lying side preference. The objective of this study was to determine the effect of lameness and hock injuries on lying behaviors, particularly lying laterality, of lactating dairy cows. Cows from four commercial farms in eastern Croatia that had lying behavior data, health score data, and production records were used in the study. Health scores including hock injuries and locomotion were collected once per cow. Severely lame cows had greater daily lying time compared to sound cows and moderately lame cows. Overall, cows spent 51.3 ± 1.2% of their daily lying time on the left side. Maximum hock score, locomotion score, hock injury laterality, or parity did not result in lying laterality differing from 50%.

## 1. Introduction

Reducing the prevalence of hock injuries and lameness is key in improving cow welfare due to the discomfort associated with these conditions and the overall high prevalence. Previous research has consistently reported substantial rates of hock injuries globally with rates ranging from 57% in France, 73% in southern British Columbia [1] to 81% in the northeastern United States [2]. The high prevalence of hock injuries indicates decreased welfare for cows experiencing severe hock injuries primarily by increasing the likelihood of cows becoming lame [3]. Although severe hock lesions increase the risk of cows developing lameness [3]; lameness may also increase the risk for the development of hock lesions [4] by increasing lying time [5] and difficulty standing and lying [6], therefore increasing exposure to the stall surface. Lameness in itself is a common problem on farms; 23.9% of cows in the United States in 2007 were lame at least once in a 12 month period [7].

The relationship between hock injuries and lameness may be due to similar environmental risk factors such as housing [1,8,9]. Freestall farms without opportunities for cattle to graze had greater incidence of lameness and knee swellings compared to freestall farms with grazing [8]. Although these conditions have similar risk factors, lying behaviors such as total daily lying time and length of lying bouts differ between lame cows and cows with hock injuries [10,11]. Lame cows have increased total daily lying time and increased lying bout durations [10], but cows with hock injuries have decreased lying bout durations [11].

Inadequate bedding and overstocking can also alter normal lying behaviors by decreasing lying time in cows [12,13]. Furthermore, the risk of lameness increases with abnormal lying behaviors and the use of mats or mattresses as a stall base [9]. Lying laterality (time spent lying on either the left or right side) may be altered by other painful conditions. Lactating dairy cattle spend approximately the same proportion of time lying on their left and right sides. However, discomfort may drive cows to favor the left side (spending 61% of lying time on the left) during pregnancy [14,15] and the right side (70% of time lying on the right side) following rumen cannulation [16]. Currently, a shift in lying laterality appears to occur as a reaction to internal or external stimuli [17], and any effect of a change in lying laterality has not been reported. Although a shift in lying laterality has not been found to occur with lameness the importance of hock injuries in the alteration of lying laterality have yet to be illustrated [18].

This study was a retrospective cross-sectional study. The aim was to determine if hock injuries (in particular, the side of hock injury) or lameness contributed to a shift in lying laterality from the expected 50% per side or an alteration in other lying behaviors. Therefore, the objective of this study was to determine the impact of hock injuries and lameness on lying behaviors, particularly lying laterality for lactating dairy cows with hock injuries and lameness. It was hypothesized that lying laterality in animals with hock injuries and/or lameness would differ from the expected 50%. Increased lying time has also been associated with higher parity [19], therefore it was also hypothesized that other lying behaviors (lying time, bouts, lying bout duration) would be altered in those animals with injuries or lameness, and lying time would increase with parity.

## 2. Materials and Methods 

Lactating Holstein dairy cows from four farms across eastern Croatia were evaluated for this study. Farms were selected due to the Memorandum of Understanding in place between the farm and the University of Osijek and their role in the Fulbright Scholarship with the intention of measuring welfare of Croatian dairy farms. A breakdown of descriptive cow data is included in Table 1. Cows were housed in either freestalls with straw bedding (farms 1, 2, and 3) or deep-bedded packs with straw (farm 4). Cows on farm 1 were milked in a 40-cow rotary parlor twice daily. Cows on farm 2 were milked in a 24 double sided herringbone parlor three times daily or twice daily for late lactation cows. Cows on farm 3 were milked using a robotic parlor system (2 robots per pen) with free choice milking. Cows on farm 4 were milked in a 20 double sided parallel parlor twice daily. Based on available freestalls stocking, density on farms 1–3 was below 100%. On the bedded pack, less than 9.3 m² was provided per cow, which is the recommended space per cow on bedded packs [20]. All heifers from all farms were raised at a common location with bedded packs and pastures. All farms fed cows twice daily and used Delaval milking equipment (Tumba, Sweden).

Lying behaviors (total, left, and right side lying time (min/d), lying bout durations (min/bout), and number of lying bouts per day (n/d)) were collected using Hobo Pendant G data loggers (Onset Computer Corp., Bourne, MA, USA) as previously validated [21] for a minimum of 3 days and summarized by day [22]. Lying behaviors were averaged by day and the mean value for the period in which data loggers were attached (minimum of 3 days) were used in the analysis. Locomotion and hock injury scoring were conducted once per cow while the data loggers recording lying behaviors were attached. Locomotion was evaluated using the National Animal Health Monitoring System (NAHMS) scoring system [23]. A sound cow was represented by a score of 1 (no gait abnormality), a moderately lame cow was represented by a score of 2 (visible gait abnormality), and a score of 3 represented a severely lame cow (prominent head bob with cow visibly favoring one or more limbs). Hocks were scored on a 0–3 scale where 0 indicated no visible injury on the hock, 1 indicated hair loss but no swelling, 2 indicated the presence of swelling (swellings smaller than 7.4 cm in diameter, no bleeding), and a score of 3 indicated major swelling (greater than 7.4 cm in diameter, may be bleeding) [24]. Right and left hocks were scored separately and recorded for all cows evaluated. Both locomotion and hock injuries were scored by one individual with extensive experience.

Cow records were obtained from each farms’ herdsman, translated from Croatian. Individual cow data included cow ID, lactation number, days in milk (DIM), pregnancy status, breeding status, days to calving, number of inseminations, days to peak milk, peak milk yield, and milk quality (% fat, % protein, somatic cell count. However, not all of the above characteristics were recorded by each farm. Therefore, only cow ID, DIM and lactation number were used for each farm, as these were the only characteristics each farm recorded.

Cows used in this study were part of a general assessment project where lying behaviors were collected from 278 cows and health scores were collected from 792 cows. For data analysis in the current study, only cows with lying behavior, health scores (locomotion and hock injury scores), and cow records (lactation number and DIM) were used (*n* = 195). Additionally, five cows that had health scores assessed twice due to being moved to a different pen were removed from the data set to keep the number of assessments consistent between cows. Power was calculated to be 0.61 (alpha = 0.05) due to the fact that only cows on farms with existing relationships with the university were available for data collection. To achieve a power of 0.80, a total of 306 cows would have been necessary. Unfortunately, the number of available cows was much lower and the small power may reduce the ability to detect effects. However, the intent of this study was to discover potential behavioral differences using the data and cows made available.

Lying behaviors (total daily lying time, daily lying bout duration, and total number of lying bouts per day) were analyzed using a mixed model (SAS, v 9.3, Cary, NC, USA) to determine effect of maximum hock score, hock injury laterality, locomotion score, and parity. Maximum hock scores of 0 and a hock injury laterality of “neither” were the same population of cows, therefore maximum hock score had to be adjusted. Maximum hock score was analyzed as maximum hock score within hock laterality. Only maximum hock scores of 1 had both bilateral and unilaterally injured groups. All cows with a maximum hock score of 2 were injured unilaterally. The experimental unit was cow and the random effect was farm. Degrees of freedom were estimated using Kenward-Roger option. One sample *t*-tests were used to determine if lying laterality (percent time spent lying on the left side) differed from 50%. Four *t*-tests were performed to determine differences in four variables of interest: maximum hock score, locomotion score, hock injury laterality, and parity. An additional *t*-test was performed to determine if lying laterality differed from 50% in the overall population of cows used in the study. In order to determine variability of lying behaviors (lying time (h/d), lying bout duration (min/bout), and bouts (n/d)) in relation to maximum hock scores, hock score laterality, locomotion scores, and parity univariate associations of SD (Table 2) were determined using a univariate analysis (general linear model procedure; SAS v 9.3, Cary, NC, USA).

Maximum hock scores were determined by using the maximum hock score from each cow. Hock scores of 2 and 3 were combined into one score (score = 2) due to the limited number of cows with a hock score of 3. Hock injury laterality was categorized into three categories: unilateral (cow has only one hock that is scored at 1 or above), bilateral (cow has both hocks scored at 1 or above), and neither (both hocks were scored as 0). Cows with only a left injury (*n* = 7) or right injury (*n* = 8) were combined into the unilateral category due to the limited number of cows with only left or right hock injuries. Dependent variables included lying time, lying bouts, and lying bout duration. Independent variables included in the model were maximum hock score, hock injury laterality, locomotion score, and parity.

## 3. Results

Severely lame cows had greater daily lying time than sound (Figure 1; *p* = 0.003) and moderately lame cows (Figure 1; *p* = 0.002). No differences in lying time were found between hock injury laterality (*p* = 0.52), max hock score (*p* = 0.79), or parity (*p* = 0.27).

Sound cows had shorter lying bout duration compared to moderately lame cows (Figure 2; *p* = 0.004) and severely lame cows (Figure 2; *p* = 0.01). Lying bout duration did not differ between hock injury laterality (*p* = 0.29), max hock score (*p* = 0.97) or parity (*p* = 0.07).

Sound cows had greater daily lying bouts than moderately lame cows (Figure 3; *p* = 0.01) but not severely lame cows (Figure 3; *p* = 0.54). Lying bouts did not differ between hock injury laterality (*p* = 0.35), max hock score (*p* = 0.85) or parity (*p* = 0.31).

Overall, cows spent 51.3 ± 1.1% daily lying time on their left side which did not differ from 50% (*p* = 0.28). The lying laterality did not differ from 50% for cows with varying maximum hock scores (*p* ≥ 0.20), locomotion scores (*p* ≥ 0.21), side of hock injury (*p* ≥ 0.20), or parity (*p* ≥ 0.14).

Standard deviation of lying time was greater in multiparous cows than primiparous cows (Table 2) but did not differ between the other variables of interest. The SD of lying bout duration decreased between unilaterally injured cows and bilaterally injured cows and unilaterally injured cows and cows without injury (Table 2). The SD of lying bout duration increased between primiparous and multiparous cows (Table 2).

## 4. Discussion

This study is the first investigation of the impact of hock injuries on lying laterality in dairy cows on Croatian farms. In addition, the impact of lameness and parity on lying laterality were investigated. The influence of these measures on lying time, bouts, and bout duration were also examined. In this study, maximum hock score, hock injury laterality, lameness, or parity did not influence lying laterality. However, lameness severity did affect lying time, bouts, and bout duration.

Severely lame cows spent more time lying down compared to sound and moderately lame cows. Similarly, previous research reports an increase in lying time in lame cows (11.1 ± 2.8 h/d) compared to sound cows (10.5 ± 2.7 h/d) [11]; although in the previous study moderate and severe lameness were not distinguished from each other. There were no differences in lying time between moderately lame cows and sound cows in the current study. Previously, moderately lame cows on mattresses and severely lame cows housed on deep bedded stalls had increased lying time compared to sound cows [10]. This suggests the effectiveness of this parameter in indicating reduced welfare in cows; therefore, it is likely that severely lame cows in the present study experienced decreased welfare compared to sound cows.

Parity did not influence lying time contrary to previous research, which reports increased lying time with increased parity [19,25]. However, the previous study separated parity by primiparous, 2nd lactation, and 3rd and greater lactation [19], where in the present study we chose to separate parity by primiparous and multiparous cows. Additionally, due to the availability of cows on farms, there were a greater number of multiparous cows (*n* = 122) than primiparous (*n* = 73). Both heifers and cows prioritize resting [26,27] and an alteration in normal lying behaviors and higher parity increase the likelihood of a cow developing lameness [9].

Overall, mean lying bout duration for multiparous (80.6 ± 30.8 min/bout), primiparous (71.5 ± 19.8 min/bout), uninjured (75.5 ± 30.9 min/bout), bilaterally injured (76.6 ± 28.4 min/bout), and unilaterally injured cows (87.2 ± 30.9 min/bout) did not exceed the 90 min/bout threshold that was found to be associated with severely lame cows [10]. This suggests that these cows were not experiencing the level of discomfort associated with severe lameness, or changes in lying behavior are expressed differently in cows with varying hock injury severities and laterality.

Sound cows had a greater number of lying bouts per day compared to moderately lame cows. This is contrary to previous data that reports no difference in lying bouts between moderately lame and sound cows [10,18]. Previous research [18] reports cows with an average of 9.8 ± 0.49 n/d and moderately lame cows with 9.4 ± 0.49 n/d compared to the 10.3 ± 0.7 n/d for sound cows and 8.9 ± 0.8 n/d for moderately lame cows in the current study. Daily lying bouts did not differ between severely lame and sound cows. Previous research [10] also reported no differences in lying bout frequency between sound cows and severely lame cows. Moderately lame cows on mattresses have increased standing time compared to sound cows and moderately lame cows on sand bedded stalls [28], which is thought to be a response to the pain related to lameness [29] and potential discomfort associated with mattresses [1]. However, the limited number of farms on the current study restricts our ability to make conclusions based on housing or bedding type.

Lying laterality did not differ between maximum hock scores, locomotion scores, parity, or hock injury laterality. Previous research reports increased right side laterality with age but not specifically with increased parity and no laterality preference in pregnant heifers [30]. In that study, laterality data was collected through live observations at 15 min intervals during seven 24 h periods with observations occurring between sixteen months [30]. Comparatively, in the present study, sampling of laterality behavior was achieved with a data logger for a minimum of 3 d at 1-min intervals. Differences in laterality may be more pronounced long term, rather than short term. Additionally, individual lying behavior differs from cow to cow [31] which would be expected to be the case for lying laterality as well. To the authors’ knowledge, no research has been published on the influence of hock injuries on lying laterality.

Evaluating SD of lying behaviors can indicate variability of lying behaviors among individual cows [32]. Variation in lying behaviors for individual cows within a herd is expected [31], but differences may be linked to negative health conditions such as hoof ulcers [5] or increased parity and age [19]. Multiparous cows had increased SD of lying time and lying bout duration compared to primiparous cows, suggesting that multiparous cows had less uniform lying behaviors than primiparous cows.

## 5. Conclusions

The current results suggest that lying laterality does not differ between varying levels of hock injury or lameness severity. However, further research could determine if a shift in lying side occurs on an individual cow level as hock injuries or lameness progress. The current study focused on a short period of time.

## Figures and Tables

**Figure 1 animals-07-00086-f001:**
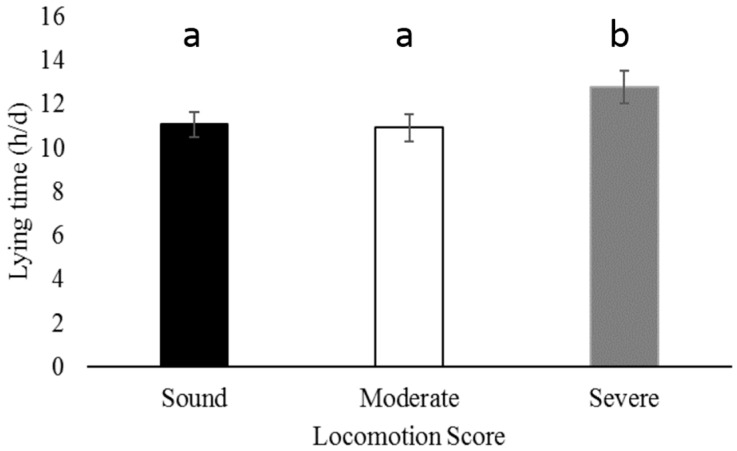
Severely lame cows (locomotion score = 3) had greater mean daily lying time than moderately lame cows (locomotion score = 2; *p* = 0.002) and sound cows (locomotion score = 1; *p* = 0.003). Differences between sound, moderately lame, and severely lame cows are designated by superscripts “a” and “b.”

**Figure 2 animals-07-00086-f002:**
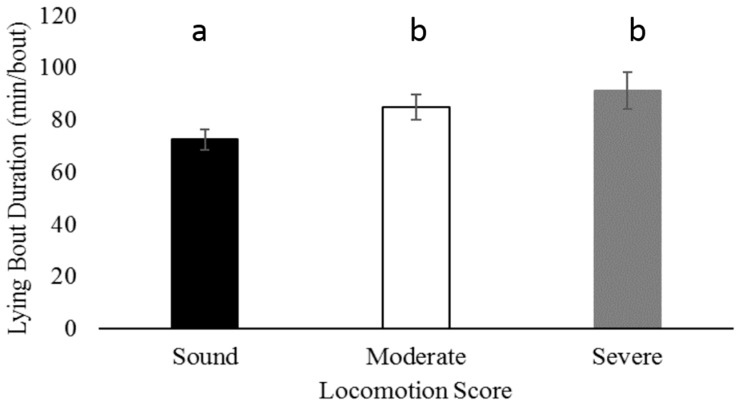
Sound cows (locomotion score = 1) had shorter mean lying bout duration compared to moderately lame cows (locomotion score = 2; *p* = 0.004) and severely lame cows (locomotion score = 3; *p* = 0.01). Differences between sound, moderately lame, and severely lame cows are designated by superscripts “a” and “b.”

**Figure 3 animals-07-00086-f003:**
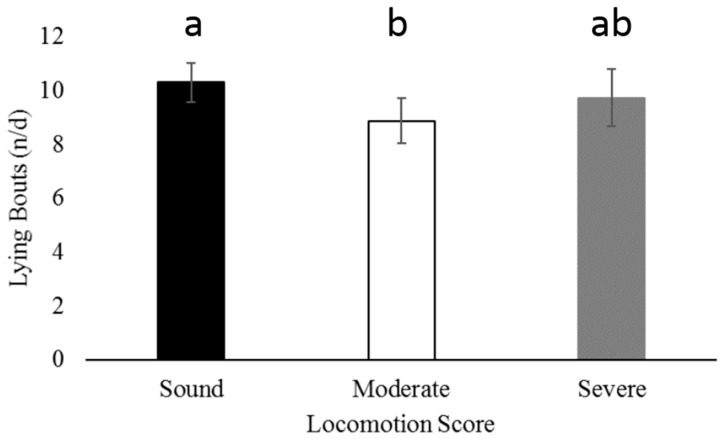
Sound cows had higher daily lying bouts compared to moderately lame cows (*p* = 0.01). Mean daily lying bouts did not differ between sound cows and severely lame cows (*p* = 0.54) or between moderately lame cows and severely lame cows (*p* = 0.37). Differences between sound, moderately lame, and severely lame cows are designated by superscripts “a” and “b.”

**Table 1 animals-07-00086-t001:** Number of cows for each variable.

Variable	Farm 1	Farm 2	Farm 3	Farm 4	Overall
Cows	32	61	42	60	195
Sound	19	42	25	31	117
Moderately lame	9	18	14	20	61
Severely lame	4	1	3	9	17
Hock score 0					
Left hock	0	15	0	31	46
Right hock	0	14	0	31	45
Hock score 1					
Left hock	15	37	23	27	102
Right hock	17	38	25	28	108
Hock score 2					
Left hock	16	6	18	2	42
Right hock	15	8	13	1	37
Hock score 3					
Left hock	1	3	1	0	5
Right hock	0	1	4	0	5
Bilateral	32	41	42	27	142
Unilateral	0	11	0	4	15
Neither	0	9	0	29	38
Multiparous	0	39	30	53	122
Primiparous	32	22	12	7	73
Early lactation	4	36	6	16	62
Mid lactation	19	25	29	32	105
Late lactation	9	0	7	12	28

Sound = locomotion score 1, no gait abnormality; Moderately lame = locomotion score 2, visible gait abnormality; Severely lame = locomotion score 3, obvious gait abnormality, cow noticeably favored one or more limbs; Hock score 0 = no visible injury; Hock score 1 = hair loss but no swelling; Hock score 2 = swelling present (but less than 7.4 cm in diameter); Hock score 3 = major swelling present (greater than 7.4 cm in diameter); Bilateral = hock injuries on both right and left hocks; Unilateral = hock injury on either the right or left hock; Neither = cow without a hock injury; Early lactation = cows less than 90 days in milk; Mid lactation = cows between 91–249 days in milk; Late lactation = cows 250 days in milk and greater.

**Table 2 animals-07-00086-t002:** Standard deviation of the herd average daily lying time (h/d) and lying bout duration (min/bout) with parity and hock injury status. Standard deviation of lying time and lying bout duration increased with multiple parities. Standard deviation of lying bout duration decreased when both hocks were injured or not injured.

Variable	Estimate	SE	R²	*p* Value
**SD of lying time (h/d)**				
Multiparous	0.31	0.13	0.30	0.02
**SD of lying duration (min/bout)**				
Bilateral injury	−14.41	4.44	0.07	0.001
No injury	−17.86	4.99	0.07	0.0004
Multiparous	5.02	2.49	0.02	0.05

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
