# Peer review of "The Effect of Hock Injury Laterality and Lameness on Lying Behaviors and Lying Laterality in Holstein Dairy Cows"

_animals, 2017, doi:10.3390/ani7110086_

Round 1

Reviewer 1 Report

Overall, the manuscript is well written and the methods applied are sound. Nevertheless, there are some issues that have to be clarified before it can be considered for publication. Overall, it does not become clear which consequences a shift in lying laterality from the expected 50% per side has. Why is it important that no shift occurs? Is there any consequence for animal health or productivity? Or is it just and indicator of welfare? Furthermore, does it really give a good indication about cow welfare? If this is the main objective of the current research authors should put more effort on explaining the underlying reason for investigating this variable.

L35: Delete “(Kester et al., 2014)“

L57: „but has not been to occur“ is grammatically incorrect. Please revise.

L90-92: This part fits better within the introduction and not within the material and methods.

L98: Please spell out “NAMHS”

L130: More information regarding the statistical analysis needed. The readers should be able to understand the applied methods without looking up further references several times.

L173: The statistical analysis needs to be explained in more details here. As also the information given in the text regarding the univariate analysis of SD is rather limited, authors should put more emphasis on this aspect in the manuscript. Without further information it is hard for the reader to follow.

Figure 1-3: Superscripts indicating significant differences are missing.

Overall, tables have to be formatted according to the guidelines provided for this journal.

Table 3: No need to present data that are already shown in graphs additionally in tables.

Table 2: This table shows results and does not give information about the applied methods, therefore it has to be shifted to the results section. Furthermore, please arrange it after table 3 in accordance with the order the results are presented in the text. Furthermore, better description of this table is needed. For instance, what does “estimate” refer to..

L214-L215: Here you state that the lying bout duration of multiparous cows compared to primiparous cows is higher, whereas in the results section you state that there was no difference. According to table 3 primiparous cows show a trend towards higher lying bout duration (87.4) compared to multiparous cows (78.6; P = 0.07). Please clarify.

L272-L75: I would be more cautious with this conclusion. Unilaterally injured cows made up a very small group (n=15) compared to bilaterally and non-injured cows (n=180). Therefore, the decreased SD in the latter group is not surprising.

L277-L286: Overall, the conclusions are too long and mainly report results. Better condense this part and report conclusions and implications of the study.

Author Response

REVIEW 2

Comments and Suggestions for Authors

Overall, the manuscript is well written and the methods applied are sound. Nevertheless, there are some issues that have to be clarified before it can be considered for publication. Overall, it does not become clear which consequences a shift in lying laterality from the expected 50% per side has. Why is it important that no shift occurs? Is there any consequence for animal health or productivity? Or is it just and indicator of welfare? Furthermore, does it really give a good indication about cow welfare? If this is the main objective of the current research authors should put more effort on explaining the underlying reason for investigating this variable.

AU: Previous research has primarily focused on a shift in lying laterality as a reaction and not the impact of the reaction. However, that would be an excellent next step. For now, the idea of a shift in lying laterality as a reaction centers around the shift occurring in response to both negative stimuli (discomfort from rumen cannulation or pregnancy) or “normal” circumstances (after eating). Therefore, one can say it may be a reaction to a current suboptimal welfare state (discomfort) but no studies (that the authors have found) go into the actual ramifications of a shift in laterality. A comment about this was added at lines 61-64.

L35: Delete “(Kester et al., 2014)“

AU: Deleted.

L57: „but has not been to occur“ is grammatically incorrect. Please revise.

AU: Sentence fixed (lines 63-64)

L90-92: This part fits better within the introduction and not within the material and methods.

AU: Moved to lines 66-68

L98: Please spell out “NAMHS”

AU: Full term added at lines 114-115

L130: More information regarding the statistical analysis needed. The readers should be able to understand the applied methods without looking up further references several times.

AU: The cited works did not provide any additional information to the statistical analysis; they were added to credit the paper for providing the idea to utilize those components. References were removed to reduce confusion.

L173: The statistical analysis needs to be explained in more details here. As also the information given in the text regarding the univariate analysis of SD is rather limited, authors should put more emphasis on this aspect in the manuscript. Without further information it is hard for the reader to follow.

AU: Additional information was provided at 153-156. More information was also provided at table 2.

Figure 1-3: Superscripts indicating significant differences are missing.

AU: Added.

Overall, tables have to be formatted according to the guidelines provided for this journal.

AU: The authors double checked the instructions to authors and added definitions of the variables in the table.

Table 3: No need to present data that are already shown in graphs additionally in tables.

AU: Table removed.

Table 2: This table shows results and does not give information about the applied methods, therefore it has to be shifted to the results section. Furthermore, please arrange it after table 3 in accordance with the order the results are presented in the text. Furthermore, better description of this table is needed. For instance, what does “estimate” refer to..

AU: Table moved and more information provided in the caption.

L214-L215: Here you state that the lying bout duration of multiparous cows compared to primiparous cows is higher, whereas in the results section you state that there was no difference. According to table 3 primiparous cows show a trend towards higher lying bout duration (87.4) compared to multiparous cows (78.6; P = 0.07). Please clarify.

AU: Statement removed. The paragraph (232-238) was adjusted based on other reviewers’ comments.  

L272-L75: I would be more cautious with this conclusion. Unilaterally injured cows made up a very small group (n=15) compared to bilaterally and non-injured cows (n=180). Therefore, the decreased SD in the latter group is not surprising.

AU: Statement removed.

L277-L286: Overall, the conclusions are too long and mainly report results. Better condense this part and report conclusions and implications of the study.

AU: Conclusions shortened.

Reviewer 2 Report

This manuscript describes a study investigating the impact of lameness and hock lesions on lying laterality, lying time, lying bouts and lying bout duration. The inclusion of lying laterality is the most novel part of the study, but requires revision before the paper can be acceptable for publication. 

A main concern with the paper is that although authors are interested in the impact of hock lesions on lying laterality, the statistical test used may not have been able to adequately answer the research question. As is, authors clumped together cows with hock lesions on the left and right into the same category (‘bilateral’), and then tested to see if all cows in that category showed a preference for left or right. A better test would be to analyze cows with hock lesions on the left and right separately to see if those with a hock lesion on the right side only would avoid lying down on the left side and vice versa. Authors are encouraged to rethink this analysis to better answer their questions.

Authors should also be clear that their interest in locomotion score and lying behavior had less to do with laterality (as laterality of lameness was not measured) and more to do with lying bouts, lying time and lying bout duration. A clarification of this in the objectives is needed.

Authors should also be careful when comparing parities, as these were not evenly distributed across farms (e.g., farm 1 had only primiparous cows, and farm 4 had mainly multiparous cows). Could any differences between parities actually be farm differences? Suggest commenting on this or removing parity from the analysis (or adding it as a covariate rather than a main effect), as this variable is less interesting than the others discussed in the paper.

Overall the paper is novel and very well written.

Line 18: Change ‘increased’ to ‘higher’ as you are comparing between groups, not within

Table 1: It is unclear from this table alone what the terms ‘Bilateral’, ‘unilateral’ or ‘neither’ means, please clarify. Also suggest defining early, mid and late lactation.

Line 98: Change NAMHS to NAHMS – you may need to spell this acronym out for the first usage

Line 98-100: Please add more details about the lameness scoring system used (e.g., how was a 0, 1, 2, etc defined?). It looks like laterality of lameness was not recorded, can authors add a sentence to justify why laterality for hock lesions were included, but not included for lameness? What were authors hypotheses about laterality and lameness in this case? Suggest revising objectives to make it clear authors were most interested in the effect of side of hock lesion on lying laterality, and the effect of locomotion on lying behaviors (not really laterality).

Line 102. Can authors add more description of the difference between swelling and major swelling?

Line 104: It is unclear how ‘health scores’ were used in the study. Suggest either adding much more details to what these health scores looked like and how they were used in the analysis, or remove them from the paper. When were health scores conducted? Was it different for each cow?

Line 115: It is unclear why cows with more than one health assessment had to be removed from the analysis, please clarify.

Line 128: change random variable to random effect.

Table 2: It is unclear how this table contributes to the paper – can authors add more details to the caption to help put this table into context for readers? A general rule is to write the caption so that readers can understand the table without the context of the paper.

Table 3. Suggest adding into the caption that the P value represents a difference from 50% in some cases (laterality), and differences between cow categories in others (lying behaviors). Authors also need to better describe what terms like ‘1 – bilateral’ etc mean in the table. When authors say ‘1-bilateral’ could that be left or right? It would be more interesting to know if cows with a hock lesion on the left spent more time lying on the right and vice versa. The way the data has been summarized would hide that effect.

Author Response

REVIEW 1

Comments and Suggestions for Authors

This manuscript describes a study investigating the impact of lameness and hock lesions on lying laterality, lying time, lying bouts and lying bout duration. The inclusion of lying laterality is the most novel part of the study, but requires revision before the paper can be acceptable for publication. 

A main concern with the paper is that although authors are interested in the impact of hock lesions on lying laterality, the statistical test used may not have been able to adequately answer the research question. As is, authors clumped together cows with hock lesions on the left and right into the same category (‘bilateral’), and then tested to see if all cows in that category showed a preference for left or right. A better test would be to analyze cows with hock lesions on the left and right separately to see if those with a hock lesion on the right side only would avoid lying down on the left side and vice versa. Authors are encouraged to rethink this analysis to better answer their questions.

AU: Initial analysis did separate cows with only right injuries or left injuries, but the authors combined them into one variable (unilateral) because there were not enough cows with only a left injury (n=7) or right injury (n=8). A statement was added to the materials and methods to reflect this (line 161-163).

Authors should also be clear that their interest in locomotion score and lying behavior had less to do with laterality (as laterality of lameness was not measured) and more to do with lying bouts, lying time and lying bout duration. A clarification of this in the objectives is needed.

AU: Although the main focus of the study was on the impact of hock injuries on lying laterality (due to the novel nature), locomotion score/lameness and lying laterality was measured. These results are reported at lines 195-196.

Authors should also be careful when comparing parities, as these were not evenly distributed across farms (e.g., farm 1 had only primiparous cows, and farm 4 had mainly multiparous cows). Could any differences between parities actually be farm differences? Suggest commenting on this or removing parity from the analysis (or adding it as a covariate rather than a main effect), as this variable is less interesting than the others discussed in the paper.

AU: A comment was added to reflect the potential issue of comparing parities at lines 235-236.

Overall the paper is novel and very well written. 

Line 18: Change ‘increased’ to ‘higher’ as you are comparing between groups, not within

AU:  Increased was changed to “greater” at line 28.

Table 1: It is unclear from this table alone what the terms ‘Bilateral’, ‘unilateral’ or ‘neither’ means, please clarify. Also suggest defining early, mid and late lactation.

AU: Definitions of each category were added for table 1.

Line 98: Change NAMHS to NAHMS – you may need to spell this acronym out for the first usage

AU: NAHMS was changed and spelled our at line 114-115.

Line 98-100: Please add more details about the lameness scoring system used (e.g., how was a 0, 1, 2, etc defined?). It looks like laterality of lameness was not recorded, can authors add a sentence to justify why laterality for hock lesions were included, but not included for lameness? What were authors hypotheses about laterality and lameness in this case? Suggest revising objectives to make it clear authors were most interested in the effect of side of hock lesion on lying laterality, and the effect of locomotion on lying behaviors (not really laterality).

AU: More information on the locomotion scoring system was added at lines 115-118. We did look at whether lame cows shifted lying laterality from 50%, so the authors would like to keep that in the objectives.

Line 102. Can authors add more description of the difference between swelling and major swelling?

AU: More information in regards to hock swelling was added at lines 119-121.

Line 104: It is unclear how ‘health scores’ were used in the study. Suggest either adding much more details to what these health scores looked like and how they were used in the analysis, or remove them from the paper. When were health scores conducted? Was it different for each cow?

AU: Clarification in regards to health scores was provided at line 132. The term “health scores” was used as a blanket term for both locomotion scores and hock injury scores, but the authors previously failed to define it.

Line 115: It is unclear why cows with more than one health assessment had to be removed from the analysis, please clarify.

AU: Clarification was added at line 135.

Line 128: change random variable to random effect.

AU: Changed (line 148).

Table 2: It is unclear how this table contributes to the paper – can authors add more details to the caption to help put this table into context for readers? A general rule is to write the caption so that readers can understand the table without the context of the paper.

AU: A more in-depth caption was added to Table 2.

Table 3. Suggest adding into the caption that the P value represents a difference from 50% in some cases (laterality), and differences between cow categories in others (lying behaviors). Authors also need to better describe what terms like ‘1 – bilateral’ etc mean in the table. When authors say ‘1-bilateral’ could that be left or right? It would be more interesting to know if cows with a hock lesion on the left spent more time lying on the right and vice versa. The way the data has been summarized would hide that effect.

AU: Other reviewers suggested removing table 3, so it was removed.

Reviewer 3 Report

This is a well constructed, albeit small retrospective study. I will suggest that the paper is accepted with the following changes:

Lines 116 - 120: The comments regarding the low power (due to sample size) would be more appropriate for the discussion, rather than the materials and methods

Line 224-225: "The study by [14] assessed farms" - either ad the authors names, or rephrase the sentence - eg "A previous study (14).... 

Author Response

REVIEW 4

Comments and Suggestions for Authors

This is a well constructed, albeit small retrospective study. I will suggest that the paper is accepted with the following changes:

Lines 116 - 120: The comments regarding the low power (due to sample size) would be more appropriate for the discussion, rather than the materials and methods

AU: The authors respectfully would like to keep the power comments in the materials in methods to provide context to the total number of cows used in the study.

Line 224-225: "The study by [14] assessed farms" - either ad the authors names, or rephrase the sentence - eg "A previous study (14).... 

AU: Sentence was adjusted, but the paragraph was removed due to other reviewers’ comments.

Reviewer 4 Report

General comments.

My main issue is the timing is not clear. You have health scores, locomotion scores and hock scores. When were these taken relative to the data collection on lying? Were all the data recorded at the same time? This needs to be much clearer.

The discussion was over long and was principally a list of we found this while others found this. There was only limited discussion of the results and what they mean

Specific comments:

25: Improved or worsened compared to what. This is not a useful sentence it simply states that if housing is bad you can make it better and if it’s good you can make it worse. The reference used is a non-peer reviewed industry website which can easily be replaced by peer-reviewed sources.  The second sentence is correct and has a good source but again it’s not relevant to the study which is not about improving hock lesions it’s about their impact. I recommend dropping the first two sentences

33 For this section it’s not ‘rather than vice versa” it’s ‘although’. Severe hock lesions increase lameness and lameness increase hock lesions. They are complementary not antagonistic

47 This is an orphan sentence – why are you suddenly talking about parity

55 cannulation of what?

56 you’ve just said this – drop the repetition

57 shown to occur

61 all lame or cows with hock injuries will be lame or injured so the last part of this sentence can be removed

63 This would be the right spot to put the comment about parity and lying time

76 How much space was there per cow? You state less than – how much less than

116 What were the assumptions of the power analysis?

Figure 1 No superscripts – but I don’t think you need them

127: Confirm max score 2 were bilateral (were both 2 or were there also 2, 1 s

Table 3: No numbers in each group

Author Response

REVIEW 3

Comments and Suggestions for Authors

General comments.

My main issue is the timing is not clear. You have health scores, locomotion scores and hock scores. When were these taken relative to the data collection on lying? Were all the data recorded at the same time? This needs to be much clearer.

AU: Clarification added at 113-114 and 132. “Health score” was used as a term to encompass both locomotion scores and hock scores. Animals were assessed while data loggers were attached.

The discussion was over long and was principally a list of we found this while others found this. There was only limited discussion of the results and what they mean

AU: Discussion condensed to focus on significant and main results.

Specific comments:

25: Improved or worsened compared to what. This is not a useful sentence it simply states that if housing is bad you can make it better and if it’s good you can make it worse. The reference used is a non-peer reviewed industry website which can easily be replaced by peer-reviewed sources.  The second sentence is correct and has a good source but again it’s not relevant to the study which is not about improving hock lesions it’s about their impact. I recommend dropping the first two sentences

AU: First two sentences dropped.

33 For this section it’s not ‘rather than vice versa” it’s ‘although’. Severe hock lesions increase lameness and lameness increase hock lesions. They are complementary not antagonistic

AU: Changed to “also” at line 41.

47 This is an orphan sentence – why are you suddenly talking about parity

AU: Sentence moved to 72-74.

55 cannulation of what?

AU: Rumen cannulation clarification added at line 61.

56 you’ve just said this – drop the repetition

AU: Sentence dropped.

57 shown to occur

AU: Rephrased at line 63-64

61 all lame or cows with hock injuries will be lame or injured so the last part of this sentence can be removed

AU: The last part of the sentence was removed (at line 71).

63 This would be the right spot to put the comment about parity and lying time

AU: Moved (72-74).

76 How much space was there per cow? You state less than – how much less than

AU: Unfortunately, the most specific stocking density and space availability was recorded as <9.3 m².

116 What were the assumptions of the power analysis?

AU: Additional information was added at 135-136.

Figure 1 No superscripts – but I don’t think you need them

AU: Superscripts added at other reviewers’ requests.

127: Confirm max score 2 were bilateral (were both 2 or were there also 2, 1 s

AU: All cows with a maximum hock score of 2 were injured unilaterally. For bilaterally injured cows, we took the highest score between the two hocks. Some cows had both hocks with a score of 2, some  2 and 1.

Table 3: No numbers in each group

AU: Table 3 was removed at other reviewers’ request.